# Removing of Anionic Dye from Aqueous Solutions by Adsorption Using of Multiwalled Carbon Nanotubes and Poly (Acrylonitrile-Styrene) Impregnated with Activated Carbon

**Khamael M. Abualnaja [1], Ahmed E. Alprol [2], M. A. Abu-Saied [3], Mohamed Ashour [2] and Abdallah Tageldein Mansour [4,5,*]**

[1] Department of Chemistry, College of Science, Taif University, Taif 21944, Saudi Arabia; k.ala@tu.edu.sa
[2] National Institute of Oceanography and Fisheries (NIOF), Cairo 11516, Egypt; ah831992@gmail.com (A.E.A.); microalgae_egypt@yahoo.com (M.A.)
[3] Polymeric Materials Research Department, Advanced Technology and New Materials Research Institute, City of Scientific Research and Technological Applications (SRTA-CITY), New Borg El-Arab City 21934, Egypt; mouhamedabdelrehem@yahoo.com
[4] Animal and Fish Production Department, College of Agricultural and Food Sciences, King Faisal University, Al-Ahsa 31982, Saudi Arabia
[5] Fish and Animal Production Department, Faculty of Agriculture (Saba Basha), Alexandria University, Alexandria 21531, Egypt
\* Correspondence: amansour@kfu.edu.sa

**Abstract:** This paper presents an estimation of the adsorptive potential of multiwalled carbon nanotubes (MWCNTs) and modified poly (acrylonitrile-co-styrene) with activated carbon for the uptake of reactive red 35 (RR35) dye from aqueous solution by a batch system. MWCNT adsorbent was synthesized by encapsulation via in situ polymerization. The copolymer material of poly (acrylonitrile-styrene) P (AN-co-ST) was prepared in a ratio of 2:1 V/V by the precipitation polymerization process. The prepared composites' properties were characterized by FTIR, SEM, Raman, mean particle size (PSA), and XRD analysis. The PSA of the copolymeric material was determined to be 450.5 and 994 nm for MWCNTs and P(AN-co-St)/AC, respectively. Moreover, the influences of different factors, for example pH (2–10), adsorbents dosage (0.005–0.04 g), contact time (5–120 min), initial dye concentration (10–50 mg L$^{-1}$), and temperature (25–55 °C). The optimum values were determined to be 2 and 4 pH, 10 mg L$^{-1}$ of RR35 dye, and 0.04 g of adsorbents at early contact time. Furthermore, the adsorption isotherm was studied using Langmuir, Freundlich, Tempkin, and Halsey models. Maximum capacity $q_{max}$ for MWCNTs and P (AN-co-St)/AC was 256.41 and 30.30 mg g$^{-1}$, respectively. The investigational kinetic study was appropriated well via a pseudo second-order model with a correlation coefficient around 0.99. Thermodynamic study displayed that the removal of RR35 is exothermic, a spontaneous and physisorption system. The adsorption efficiency reduced to around 54–55% of the RR35 after four cycles of reuse of the adsorbents at 120 min.

**Keywords:** water treatment; multiwalled carbon nanotubes; copolymer; reactive red 35; adsorption process

## 1. Introduction

Water pollution is a major issue faced by the world today [1]. Industrial wastewater contains many dangerous and virulent pollutants that severely affect the ecosystem [2,3]. Among the several industrial pollutants, dye-colored organic materials represent one of the essential classes of pollutants [4]. Dye contains various contaminants, including disintegrated solids, alkaline, acids, harmful pollutants, and color; numerous dyes are toxic to some living organisms [5–7]. Releasing dyes into the hydrosphere usually results in environmental damage, as colors give water annihilation of their motivational capabilities

and diminish daylight penetration, in addition to the fact that some dyes are poisonous to the environment. To minimize the danger of pollution produced by such effluents, these effluents should be treated before being released into the environment [8,9]. There are several treatment technologies used to decolorization dye from wastewater, for example adsorption, chemical coagulation, precipitation, membrane filtration, biodegradation, active sludge, oxidation, and biological methods [10]. Among these methods, adsorption is quite possibly the most efficient technique, and it is generally utilized since different methods require an enormous amount of chemicals compounds and high energy, as well as being costly. This technique transfers dyes from wastewater to a solid stage. Numerous engineered and common polymers are utilized as adsorbents. Modern researches focused on applying the nanomaterials as adsorbents for the removal of dyes from aqueous solution [11,12]. Multiwalled carbon nanotube (MWCNT) materials were used for the effective removal of dyes from aqueous effluents. MWCNTs represent a promising alternative to other absorbents for the elimination of dyes pollutants from wastewater effluents since they have large specific surface area, distinct and modifiable surface chemistry, in addition to being hollow, small size and layered structures [13]. MWCNTs have a great possible for the elimination of dyes pollution as reported many studies as Bahgat et al. [14] and Yao et al. [15]. In addition, Mohammadi and Veisi [16] studied elimination of methylene blue dye by MWCNT from aquatic solutions. The results were found to be an excellent potential of 90.90 mg g$^{-1}$ of dye.

Recently, Zare et al. [17] used carbon nanocompound for the removal of Congo red dye in aqueous solution. The maximum capacity of adsorption process was found to be $q_e$ (352.11 mg g$^{-1}$) and 92% of Congo red. Polymeric materials of poly (acrylonitrile-styrene) with cyano groups that can be effectively changed into other functional groups were utilized for the preparation of chelating adsorbent [18]. The use of MWCNTs and poly (acrylonitrile-styrene) for dye adsorption requires further studies to achieve improved mechanical properties, thermal stability, and removal of inorganic and organic pollutants. In addition, molecular chains of P(AN-co-St) holds a cyano group, which may be improved. It can be hydrolysable and adjusted to realize functionality in a numeral of applications. It has been used for the treatment of heavy metal and dyes wastewater as reported previously by [19]. Poly acrylonitrile material was prepared by Tanyol et al. [20] then applied for removal of brilliant green dye from aqueous solution, which revealed a good performance in the treatment of dyes, with a maximum adsorption capacity recorded at the level of 23.81 mg g$^{-1}$. In addition, commercial activated carbon compounds are developable in the sense that they are suitable for several of applications. Adsorption on polymers with activated carbon is presently widely utilized for the elimination of various kinds of dyes owing to their large adsorption capacity, a suitable pore size distribution, high external area, and fast sorption kinetics [21,22].

The aim of this study is to examine the adsorption of reactive red 35 textile dye from aqueous solution onto multiwalled carbon nanotubes (MWCNTs) and poly (acrylonitrile-styrene) with activated carbon, thereby allowing for comparisons to be made. This paper explored the application of available, cheap, and highly stable copolymer in the removal of unsafe materials. Moreover, the preparation and characterization of copolymer using techniques such as FTIR, PSA, SEM, Raman, and XRD analysis, in addition to determining optimization conditions. Furthermore, the kinetics and isothermal study of adsorption were investigated.

## 2. Materials and Methods

### 2.1. Preparation of Reactive Red 35 Solution

Reactive red 35 (RR35) dye, as single azo class, has a molecular formula of $C_{12}H_{18}N_3Na_3O_{14}S_4$, a molecular weight of 732.94 g mol$^{-1}$, and a wavelength of $\lambda$ max = 511 nm (Figure 1). The stock solutions of reactive red 35 were prepared in distilled water. The dye was utilized as an adsorbate without of any cleaning. The absorption of the color of

the RR35 dye was measured via a UV/VIS spectrophotometer system (Milton Roy, spectronic 2ID) on a wavelength of 410 nm. The adsorbed quantity of RR35 dye was expressed by a standard calibration curve.

**Figure 1.** Chemical structure of reactive red 35 dye.

## 2.2. Preparation of the MWCNT and P(AN-co-ST) Adsorbents

### 2.2.1. Synthesis of MWCNTs

MWCNTs were produced by the chemical vapor deposition technique, in which acetylene with cobalt and an iron solution were placed in an inert gas atmosphere connected to a reaction chamber. In this process, nanotubes were made on the substrate through the decay of hydrocarbon at an atmospheric pressure of from 600 to 900 °C. This method can be scaled up to prepare industrial amounts of MWCNT according to Bahgat et al. [14] and Eldeeb et al. [23], who reported the synthesis of MWCNT with the following steps: two grams of the supporting catalyst was injected into an alumina boat and displayed on the cylindrical quartz tube fitted inside a furnace at 600 °C, and the catalyst was heated in the presence of $N_2$ gas for 10 min at a rate of 90 mL min$^{-1}$. Movement of acetylene gas was tolerated by the quartz tube undergoing catalyzation at 90 mL min$^{-1}$ with a flow rate of 40 min after the catalyst was heated. Acetylene flow ceased after the required period, and the new product was cooled at room temperature.

### 2.2.2. Purification and Functionalization of MWCNTs

The enormous surface area prompts a strong tendency to shape agglomerates. Surface functionalization helps in steadying the scattering, since it can repress reaggregation of nanotubes and can prompt the coupling of polymeric grids with MWCNTs. Covalent functionalization of MWCNTs can be determined by identifying some functional groups on sites of MWCNTs through oxidizing agents, for example strong acids, which causes the formation of hydroxyl groups or carboxylic (–OH, –COOH) on the outside of the nanotubes. Such groups are known as functionalization-type defect groups [14]. The functionalization process was performed as reported by Eldeeb et al. [23]: 10 mL of concentrated sulfuric acid and 30 mL of nitric acid were placed in a 250 mL flask loaded with 10 g of the produced MWCNTs and 5 g phosphorous pentoxide. The mixture solution was refluxed for 120 min at 350 °C to obtain an MWCNT suspension mixture. Then, the mixture solution was washed with deionized water followed by drying for 24 h at 50 °C to obtain carboxylate MWCNTs.

### 2.2.3. Synthesis of P(AN-co-ST) and Activated Carbon Copolymer Nanoparticles

Firstly, particles from the P(AN-co-ST) copolymer were prepared by simple precipitation polymerization. The copolymerization method took place using the co-solvent from ethanol and distilled water as a solvent followed by injection of initiator (0.01 M) potassium persulfate ($K_2S_2O_8$) at 55 °C for 4 h. Secondly, after completing this process, the

polymer was separated via centrifugation at 14,000 rpm (high speed) and then washed numerous times with an ethanol-distilled water solution to remove any unreacted monomers or excess initiator. Thirdly, the white polymeric product was dried in an oven at 70 °C overnight. Finally, the white polymeric product P (AN-co-ST) was mixed with commercially activated carbon with a ratio of 50:50 [24].

### 2.3. Adsorption Experiments

Adsorption studies for the estimation of the capabilities of both MWCNTs and P(AN-co-St) mixed with activated carbon adsorbents to eliminate RR35 dye from aqueous solutions were conducted following a batch adsorption process under various parameters. For these experiments, several initial RR35 concentrations (10, 20, 30, 40, and 50 mg L$^{-1}$) were mixed with various dosages (0.005, 0.01, 0.02, 0.03, and 0.04 g), contact times (15, 30, 45, 60, and 120 min), and initial pH levels of dye (2, 4, 6, 8, and 10), with all these conditions under room temperature (25 ± 2 °C). In addition, different temperatures ranges (25, 35, 45, and 55 °C) were adjusted to test the effect of temperature. For each run, dye stock solutions were prepared, certain amounts of dye solution (20 mL) were added to a 50 mL conical flask, and shaking in a platform shaker at 130 rpm for a specific amount of time was performed. Then, the solution in all flasks was filtered using filtered paper, and the last concentration of RR35 within the filtrate was measured using a UV/vis spectrophotometer.

Regeneration of adsorbent compounds was carried out after absorbing RR35 with the reusability of the adsorbent material. The used compound samples were soaked in distilled water, dried, and then shaken for 2 h. The experiment was tested using RR35 dyes at 0.01 g of composites; pH = 7; an agitation speed of 120 rpm; 20, 30, 40, and 50 mg L$^{-1}$ dye concentrations.

### 2.4. Characterization of MWNT and P (AN-co-St)/AC

The morphologies of P (AN-co-St) with activated carbon and MWNT were characterized by using a scanning electron microscope (JEOL JSM 6360 LA). The samples were assessed using a particle size analyzer (Beckman Coulter, USA) at an angle of 11.1° to determine the distribution of particle size. Fourier transform infrared and Raman analysis spectrophotometry used to measure the influence of the prepared polymeric materials were performed using a Shimadzu FTIR-8400 S (Japan) and a Bruker Senterra Raman spectrometer, respectively. X-ray diffraction apparatus (D8 Advance X-Ray Di-ractometer, USA) was used to investigate the crystal structure of MWCNTs with Panalytical in the scope of from 10 to 90° with K$\alpha$ Cu radiation. The measurement of the surface area of the MWCNTs was investigated by the Brunauer–Emmett–Teller model, which is dependent on the absorption of N$_2$ gas.

### 2.5. Dye Removal Efficiency

The removal percentage of RR35 dye is expressed by the following relationship [25]:

$$\text{Percentage removal (\%)} = \frac{(C_i - C_f)}{C_i} \times 100 \tag{1}$$

Where $C_i$ and $C_f$ (mg L$^{-1}$) are the first concentration and equilibrium concentration of RR35 dye, respectively. The equilibrium sorption capacity ($q_e$) was evaluated using the following Equation [26]:

$$q_e = \frac{(C_i - C_f) \times V}{m} \tag{2}$$

Where $q_e$ is the equilibrium sorption capacity (mg g$^{-1}$), V is the volume of the mixture (in liters), and m is the mass of the nanocompound applied (in grams).

## 2.6. Adsorption Equilibrium Isotherm

### 2.6.1. Langmuir Model

The isothermal models were selected taking into account that the Langmuir model shows monolayer sorption on the external surface of the adsorbent compound based on the assumption that the intermolecular powers decrease quickly with distance and predict the presence of monolayer coverage of the sorbate at its external surface [27].

### 2.6.2. Freundlich Model

The Freundlich model indicates that the retention of dye ions occurs in numerous layers, and it is employed to demonstrate heterogeneous systems and describe reversible sorption processes [28].

### 2.6.3. Tempkin Isotherm

The Tempkin model considers the heat of sorption process reductions linearly through the adsorption coverage due to adsorbate–adsorbent interactions [29].

### 2.6.4. The Halsey Model

The Halsey isotherm model is suitable for multilayer adsorption, and the fitting of the Halsey equation can be applied to heteroporous solids [30]. The mathematical equations of the Langmuir, Freundlich, Tempkin, and Halsey isotherm models can be applied in some equations that are presented in Table 1.

**Table 1.** Non-linear and linearized forms of isothermal models.

| Isotherm | Nonlinear | Parameters | Ref. |
|---|---|---|---|
| Langmuir | $$q_e = \frac{Q_m K_a C_e}{1 + K_a C_e}$$ $$\frac{1}{Q_e} = \frac{1}{bq_{max}} \times \frac{1}{C_e} + \frac{1}{q_{max}}$$ $$R_L = 1 / (1 + bC_i)$$ | $q_e$: capacity of adsorption at equilibrium; $q_{max}$: maximum sorption capacity, b: Langmuir constant. $R_L$: the dimensionless equilibrium parameter | [27] |
| Freundlich | $$q_e = K_f C_e^{1/n}$$ $$\log q_e = \log K_f + \frac{1}{n} \log C_e$$ | $K_f$: Freundlich constant; n: adsorption intensity | [28] |
| Halsey | $$\text{Ln } q_e = \frac{1}{n} \text{Ln } K + \frac{1}{n} \text{Ln } Ce$$ | n and K are Halsey constants. | [30] |
| Tempkin | $$q_e = B \ln A + B \ln C_e$$ | B is Tempkin constant = (RT/b, J mol$^{-1}$) and related to the heat of adsorption, which T is absolute temperature (K); R (8.314 J/mol K) is the ideal gas constant, and A is the equilibrium binding constant (L min$^{-1}$) related to the higher binding energy. | [31] |

## 3. Results and Discussion

### 3.1. Characterization of the MWCNTs and Modified P (AN-co-St)/AC

#### 3.1.1. FTIR Examination

The FTIR technique was applied to identify the groups responsible for dye adsorption and to study the external groups of the adsorbents (MWCNT and P (AN-co-St)/AC). FTIR spectroscopy of the functionalized MWCNT composites is presented in Figure 2A. In the spectra of the functionalized MWCNT material, the peaks at 3510–3780 cm$^{-1}$ represent N–H and O–H stretching of the carboxylic group, respectively [32]. The peak at 3075 cm$^{-1}$ is due to the N–H group [18,28], while the band at 2358 cm$^{-1}$ is referred to as thiol S-

H stretching. In addition, the presence of the band at 1654 cm$^{-1}$ is attributed to the corresponding C=N due to the reaction with hydroxyl amine. Furthermore, there are corresponding C–H groups to the peaks at 1333 cm$^{-1}$. The band at 1148 cm$^{-1}$ is attributed to C–O stretching. The bands around 453–992 cm$^{-1}$ are caused by the stretching vibration of C–H.

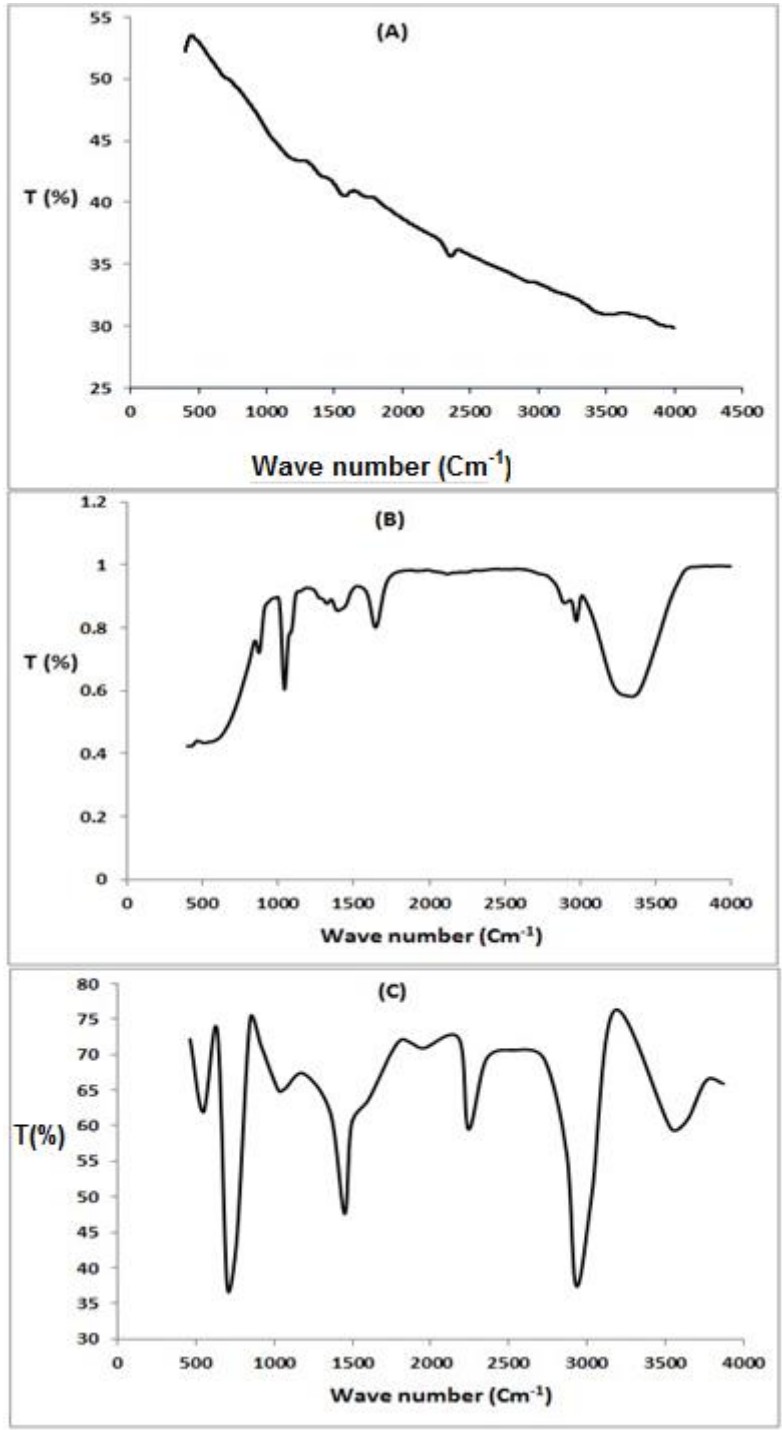

**Figure 2.** FTIR spectra of MWCNT (**A**), P (AN-co-St)/AC (**B**), and P (AN-co-St)(**C**).

Furthermore, the spectra of P(AN-co-St)/AC are presented in Figure 2B; there is a covering on the O–H and N-H stretching vibrations with a peak at 3173–3759 cm$^{-1}$. The bands at 2531 cm$^{-1}$ are attributed to the thiol S–H stretching vibration, and the band at 2929 cm$^{-1}$ was attributed to that of C–H [31]. Moreover, there is peak at 2240 cm$^{-1}$ that is

attributed to C≡N of acrylonitrile due to the reaction of additional (C≡N) groups with hydroxyl amine [2]. In addition, the shifting of the band at 1746 cm$^{-1}$ indicates C=C. The peak at 1601 cm$^{-1}$ is attributed to N−H bending, and the bands at 1080–1069 cm$^{-1}$ are due to C=O stretching. The peaks at 604–931 cm$^{-1}$ are as a result of C−H stretching vibration.

Figure 2C illustrates the spectral property peaks of the functionalized P (AN-co-St) nanocomposite. The intense absorption peak at 1449 cm$^{-1}$ is specific of C–C stretching, and the two bands at 846 and 700.1 cm$^{-1}$ are definite to C–H stretching of the aromatic ring. Attendance of a peak at 1185.3 cm$^{-1}$ is because of aliphatic C–O stretching. Furthermore, the bands in the area at 1333–1552 cm$^{-1}$ are assigned to the NH$_2$. While, the peak at 1068 cm$^{-1}$ corresponds to the C–N stretching of RNH$_2$, and the sharp band at 700 cm$^{-1}$ is attributed to the styrene ring modes [17]. Existence of the peak at 2240 cm$^{-1}$ corresponds to aliphatic C≡N stretching of pure acrylonitrile [2]. Additionally, the band that emerged at 1449 cm$^{-1}$ is characteristic of C–C stretching. Moreover, the signal at 1747.5 cm$^{-1}$ is corresponding to the carbonyl group (C=O) [33].

By the functional groups present in MWCNTs and P(AN-co-St)/AC, the interactions of the adsorbents with the RR35 dye would happen by O–H, C≡N, carboxylate NH$_2$, aromatic rings, thiol S–H, and C=O, as previously shown for the interaction of dyes with both adsorbents [33]. Additionally, Maryam et al. [34] nanotubes owing to their carbonic nature and as well existence of van der Waals attraction among tubes are hydrophobic; therefore, they display low dispensability in water and organic solvents and irradiating them with ultrasonication does not increase their dispersibility.

### 3.1.2. Particle Size Distribution Analysis (PSD)

Analysis of particle size distribution of three compounds presented in Table 2 shows that nanocomposites have a positive effect on the size of the mean particles, which were 994.0, 450.5 and 56.6 nm for MWCNTs, P (AN-co-St)/AC and P (AN-co-St), respectively, with samples reacting to an angle of 11.1°.

**Table 2.** Particle size distribution analysis of MWCNTs, P (AN-co-St)/AC, and P (AN-co-St).

| Nanocomposite | Angle | Mean (nm) |
|---|---|---|
| MWCNT | 11.1° | 994.0 |
| P (AN-co-St)/AC | 11.1° | 450.5 |
| P (AN-co-St) | 11.1° | 56.6 |

### 3.1.3. SEM Examination

This method is useful way of examining the surface morphology of MWCNTs and P (AN:ST)/AC. Figure 3A and 3B exhibit SEM photographs of P (AN-co-St) (A) and P (AN-co-St)/AC (B). These figures show the spherical, uniform shape structure pf P (AN-co-ST) and the rough structure of modified P (AN-co-St)/AC particles with an average size 450.5 nm compared with the average size of P (AN-co-St) particles (56.6 nm), showing that the modification method caused an increase in the modified composite particle size by 7.95% of its size. Furthermore, Figure 3C displays the SEM pictures of prepared MWCNTs. This figure shows the presence of amorphous carbon in the untreated sample. Additionally, it is noticeable that the tubes are thin, long, and did not agglomerate together a lot. The tubular diameter of the MWCNTs ranged between 75 and 99.25 nm, and the wall thickness fluctuated from 7.9 to 10 nm.

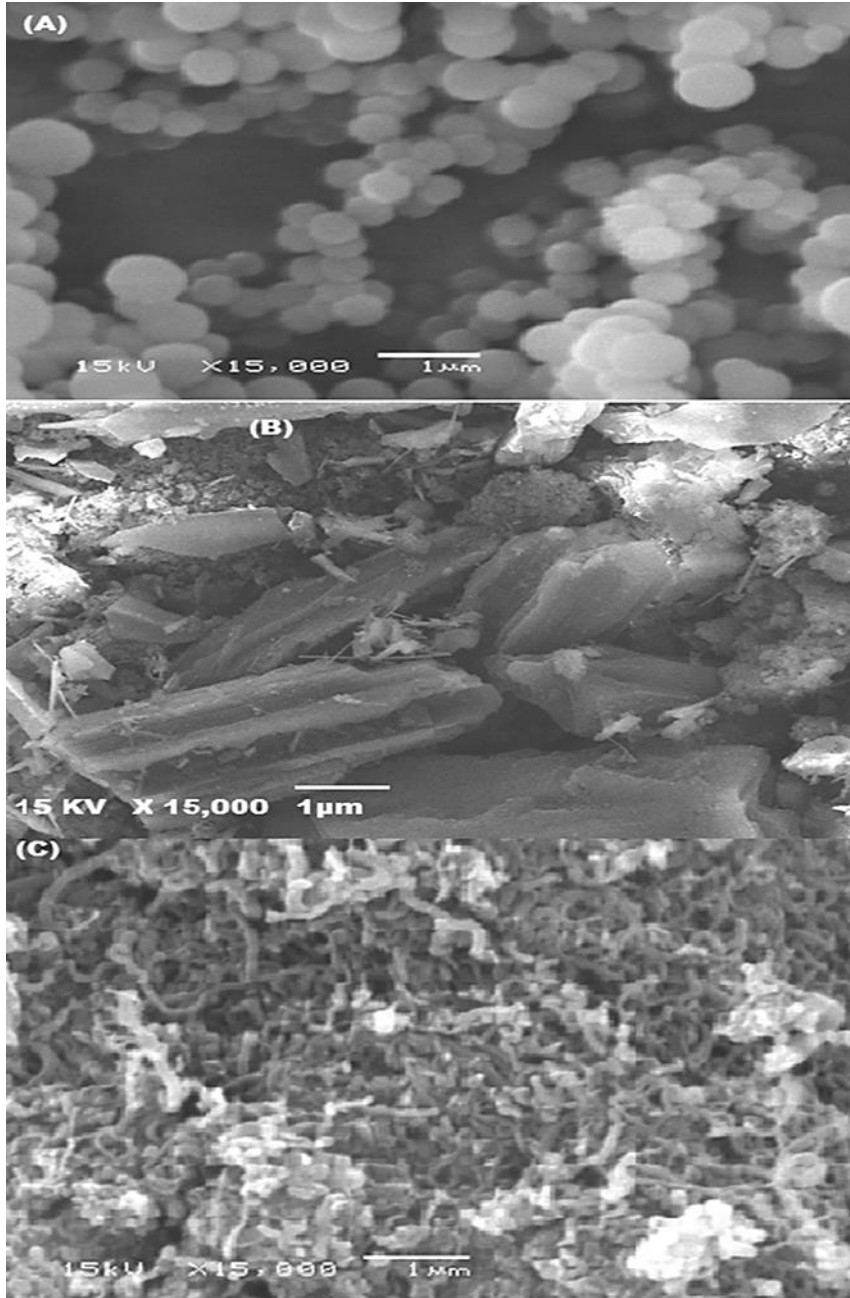

**Figure 3.** SEM images of P (AN-co-St) (**A**), P (AN-co-ST)/AC (**B**), and prepared MWCNTs (**C**).

### 3.1.4. Raman Spectral Analysis

Raman spectroscopy is usually used because vibrational information is characteristic of the symmetry of particles and chemical bonds through which molecules can be identified. Raman spectra of the P (AN-co-St), P (AN-co-St)/AC, and MWCNT after the adsorption of dye as shown in Figure 4 (A–C). The beaks intensity fluctuating from ~ 1300–1350 cm$^{-1}$ characterized the disordered carbon (D band) with Raman shift 1310, 1346.5, and 1320 for P (AN-co-St), P (AN-co-St)/AC, and MWCNT, respectively, prompted by sp$^3$ electronic states (deliberated to be defects in the planar sp$^2$ graphitic structure) were visualized. Furthermore, the bands ranging from 1580 to 1611 cm$^{-1}$ indicated the graphitic carbon G band. G band is correlated to the graphite E$_2$g symmetry of the interlayer manner. This manner reveals the structural integrity of sp$^2$-hybridised carbon atoms of the nanotubes. Together, these bands D besides G can be applied to calculate the extent of carbon-comprising defects [32]. The D peak for P (AN-co- St) was ~ 1320 cm$^{-1}$, while the G peaks

were at ~ 1599.5 cm⁻¹. The ratio of ID/IG was calculated to estimate the variation of MWCNTs quality and quantity of different defects. The ratio for both P (AN-co-St) and MWCNTs was 0.819 and 0.839, respectively. A smaller ID/IG ratio (<1.0) of MWCNTs confirmed the high quality of the samples, practically without defects or amorphous carbon. The sample has quite a great numeral of defects if the intensity of these bands is proportional [35].

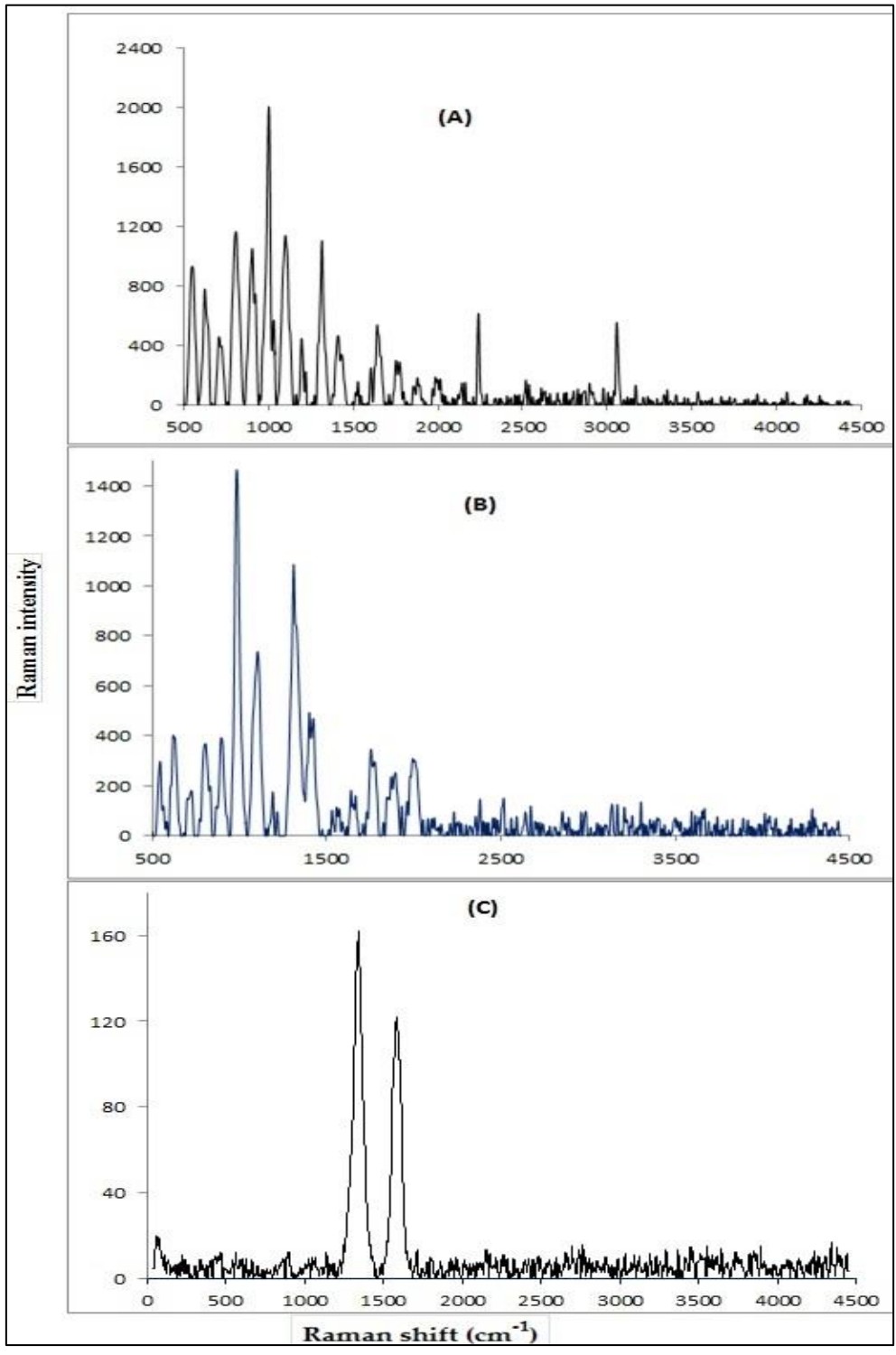

**Figure 4.** Raman analysis of P (AN-co-St) (**A**), P (AN-co-St)/AC (**B**), and MWCNTs (**C**).

### 3.1.5. XRD Analysis of MWCNTs

From the XRD pattern of the MWCNT sample displayed in Figure 5, we can conclude that there are strong bands at 2θ at 27.9, 31.28, 34.06, and 44.66° for the oxidized MWCNTs. This could be due to (002) and (100) planes of cleaned MWCNTs (graphite structure) providing improved crystallinity and a decline in the amorphous structure as a result of acid treatment [36].

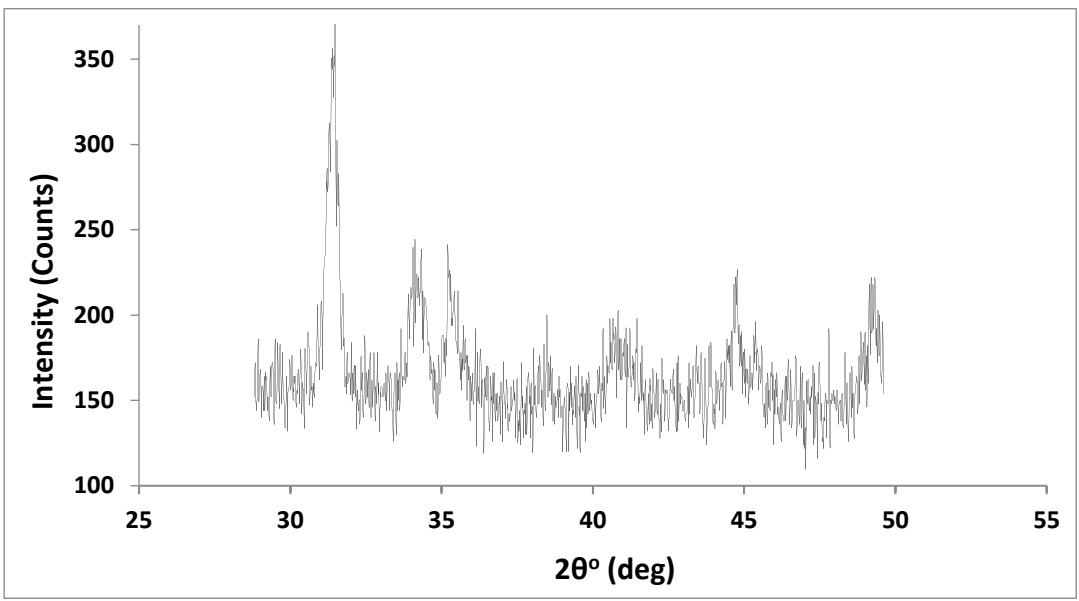

**Figure 5.** X-ray diffraction of MWCNTs**.**

### 3.1.6. BET Analysis

The specific external area and mean pore diameter of the MWCNTs were investigated by means of $N_2$ desorption–adsorption measurements at temperature 77 K as adsorption temperature and a solution $N_2$ 89.62 kPa saturated vapor pressure. The pore volume was determined from the quantity of $N_2$ adsorbed, and the pore diameter was shown to be 6.56 nm in this work. Furthermore, the results of the BET surface area examination of MWCNTs show that the whole surface area was observed to be 180.89 $m^2$ $g^{-1}$. It was shown that the MWCNTs have a greater specific surface area than that distinguished by Hanbali et al. [37]. The specific surface area of MWCNTs is affected by their agglomeration, attendance of impurities, diameter, surface functionalization, etc. [38].

### 3.2. Effect of Adsorption Conditions on RR35 Dye Adsorption

### 3.2.1. Influence of pH on Dye Adsorption

The pH of the aqueous solution is a significant factor that influences the dye adsorption process by changing the ionization conduct of adsorbent [39,40] and dye in addition to surface charge of the adsorbent. The efficiency of both MWCNTs and P (AN-co-St)/AC composites on RR35 uptake at various pH levels is presented in Figure 6. It can be seen that reactive red 35 dye elimination efficiencies increased with an increase in pH scope of 2–6, and that there was no increase in adsorption beyond this pH range. The maximum removal was achieved at pH levels of 2 and 4 with removal percentages of 95.57 and 98.41% for both MWCNTs and P (AN-co-St)/AC composites, respectively. This is due to the fact that the sorption capacity is highly pH dependent and because the adsorption mechanism relies upon the adsorbent surface charge and the ions of RR35 species in the solution. It can be noted that at a low value of pH, the binding capacity of the RR35 dye increased due to the anionic characteristics of the dye [41]. In addition, the surfaces of the MWCNTs display an anionic group (COO⁻) and subsequently improve their particle

exchange properties for particular treatments of oppositely charged species (dyes) from aqueous mixture via the electrostatic power of their interaction [42]. In the acidic condition (pH = 2 and 6), electrostatic attraction between the surface of poly multiwalled carbon nanotubes and dye particles [12] causes effective adsorption of RR35. Machado et al. [33] examined the adsorption of reactive red M-2BE dye on carbon nanotubes and activated carbon, and the removal of this dye was achieved at pH 2. Additionally, Konicki et al. [43] found that at acidic pH conditions, a positively charged external sites on MMWCNTs-ICN favor the sorption of dye anions owing to the electrostatic attraction, and the pH of the dye solution augments, the number of positively charged sites on MMWCNTs-ICN adsorbent decreases, and the number of negatively charged sites increases.

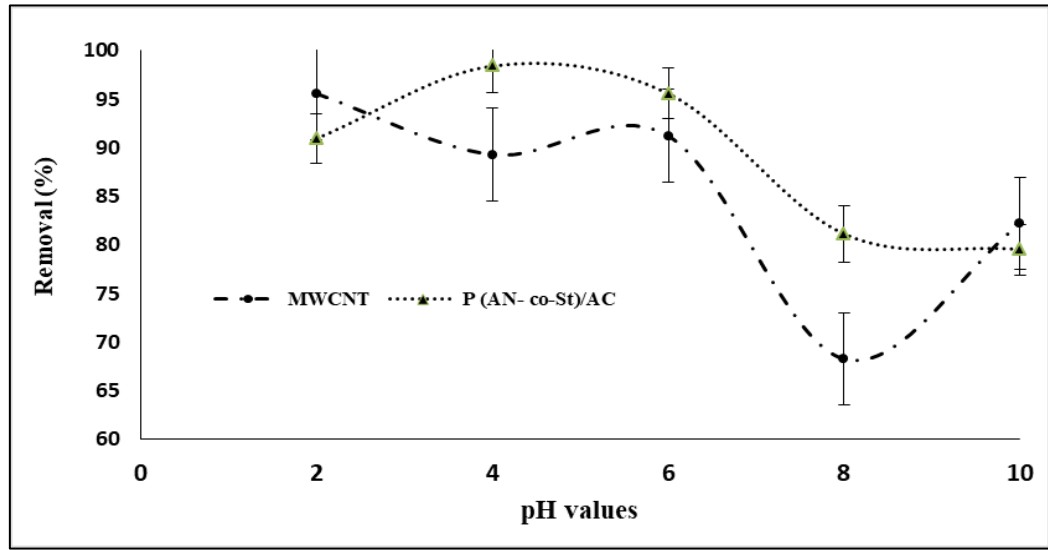

**Figure 6.** Influence of pH values on sorption of reactive red 35 (RR35) dye.

### 3.2.2. Effect of MWCNT and P (AN-co-St)/AC Composite Dosage

The amount of adsorbents is an important factor that plays a huge role in the adsorption process. To determine the ideal dose of the adsorbent, amounts of MWCNT and P (AN-co-St)/AC were varied from 0.005 to 0.04 g. Figure 7 shows the removal percentages (%) and different doses of MWCNT and P (AN-co-St)/AC. It was observed that the amount of dye adsorbed onto the unit weight of the sorbent increased from 0.005 to 0.04 g. The maximum removal was observed at 0.04 g with percentages of 97.5 and 92.3% by P (AN-co-St)/AC and MWCNT, respectively. Consequently, it can be inferred that the increase in the adsorbent amount enhances RR35 removal. The increase in the percentage of color removal is due to the existence of a large surface area with an increasing MWCNTs and poly (St-co-AN)/AC dose, which in turn causes an increase in the reactive vacant adsorption sites creation, making it easier and increasing the ionic interactions. These sites favor the transfer of RR35 solute to the outside surface of the sorbent and the increase in its elimination rate, as reported by Ghaedi et al. [44].

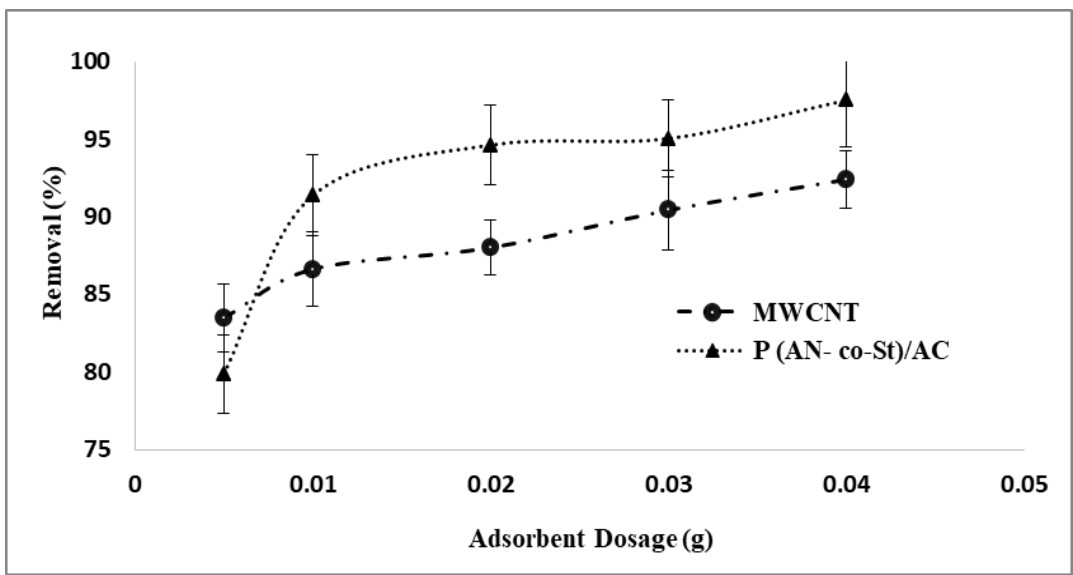

**Figure 7.** Influence of adsorbent dosage on the removal technique.

### 3.2.3. Influence of Initial Dye Concentration

The influence of the various initial dye concentrations on RR35 dye uptake by MWCNT and P (AN-co-St)/AC composites was examined at various concentrations (10–50 mg L$^{-1}$). From the achieved results presented in Figure 8, it can be observed that the removal of RR35 from two compounds is dependent on the initial dye concentration, increasing with a decrease in this concentration and decreasing at a higher initial concentration. The results showed that the amount of dye adsorbed per unit mass of adsorbent increased from 18.30 to 78.73 mg g$^{-1}$ for MWCNT, while for P (AN-co-St)/AC, it rose from 18.26 to 88.75 mg g$^{-1}$ by increasing the initial dye concentration from 10 to 50 mg L$^{-1}$. This is in agreement with results reported by other studies [41,43,45,46]. Additionally, the data showed that the increase at the initial dye concentration improves the quantity of RR35 dye adsorbed onto the compounds. The increase in the adsorption capacity is possibly owing to greater interaction among the dye and adsorbent in addition to an increase in the number of dye particle collisions that increase the driving force of the concentration gradient with the increase in the initial concentration to overcome all resistances of the dye mass transfer among the solid phases and liquid [47,48]. Moreover, the higher amount of dye adsorption at higher concentrations is probably because of increased diffusion and decreased resistance to dye uptake [49]. As well as this, the maximum removal percentage was achieved at an initial concentration of 10 mg L$^{-1}$, with removal percentages of 92 and 92.6% for both MWCNT and P (AN-co-St)/AC adsorbents, respectively. This can be explained by the fact that in RR35 adsorption process, dye concentration molecules first encounter the boundary layer effect; following this, they must diffuse from the boundary layer film onto the adsorbent surface; finally, they must diffuse into the porous structure of the adsorbent. Notwithstanding, the increase in the concentration of dye results in a reduction in the early rate of outside diffusion and expansion in the intraparticle diffusion rate [26].

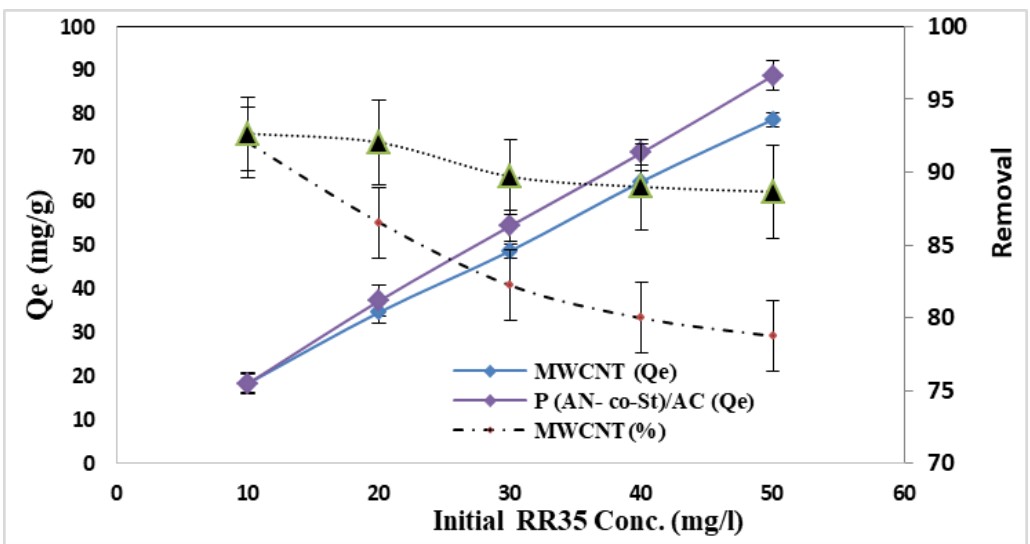

**Figure 8.** Influence of initial RR35 dye concentration.

### 3.2.4. Effect of Adsorption Time

Figure 9 shows the influence of adsorption time by utilizing MWCNT and P (AN-co-St)/AC compounds containing 0.01 g of adsorbents. From the plot, we note that the elimination of color of RR35 increased sharply in the initial 5 min and then increased slowly until it reached removal levels in a period of 1 h in the case of MWCNT and P (AN-co-St)/AC. Adsorption time increased from 59.54 to 84.30% and augmented from 19.48 to 89.53% with an increase in contact time from 5 to 60 min for both MWCNT and P (AN-co-St)/AC, respectively. In addition, beyond the attained, almost constant value, there was no notable increase in color removal. This is because the adsorbent has limited adsorbent sites, and, after a certain time, these are exhausted and the adsorption process attains an equilibrium state[50]. In addition, it is also a consequence of the development of monolayer coverage on the external surface of the adsorbent [51]. The slow rate of RR35 sorption after the first one hour possibly arose owing to the slow pore diffusion of ions of the solute into the majority of the sorbent [10].

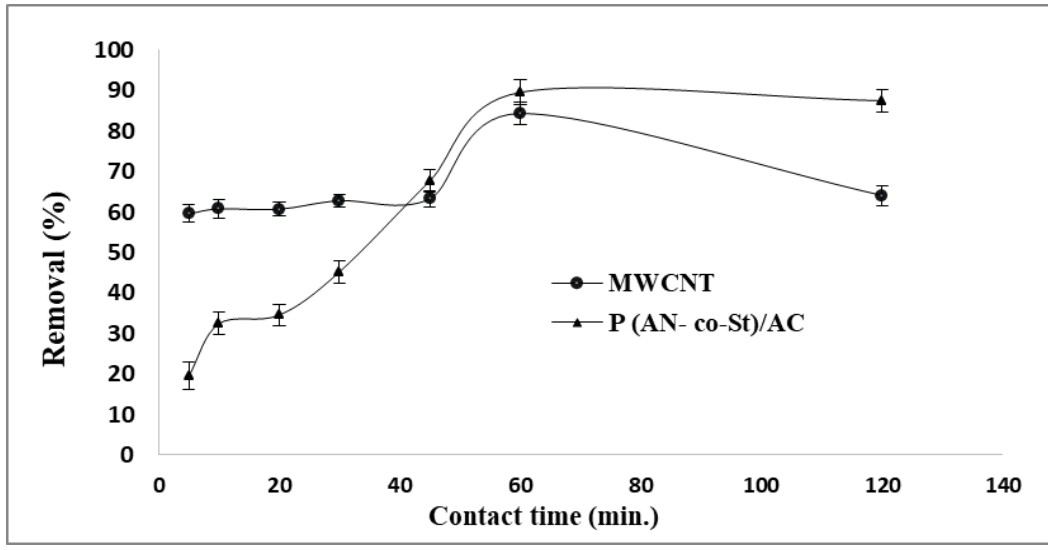

**Figure 9.** Influence of contact time on removal of RR35.

### 3.2.5. Influence of Temperature

The influence of temperature on the adsorptions of RR35 onto MWCNTs and P (AN-co-St)/AC was examined at 25, 35, 45, and 55 °C, while keeping other parameters constant for the removal of RR35 dye, as shown in Figure 10. This figure demonstrates that the adsorption process of RR35 by MWCNTs decreased with increasing temperature from 25 °C (63.33%) to 55 °C (9.07%), indicating that the processes are exothermic. Because the adsorptive powers among adsorbate and active sites on the adsorbent became weak with the increase in temperature, dye removal efficiency decreased [52]. The adsorption of the P(AN-co-St)/AC adsorbent increased with the increase in temperature of the mixture from 25 °C (67.55%) to 55 °C (97.61%). This is due to expanded surface coverage and because the sites were made reactive and active. Higher temperatures accelerate the reaction rate and diminish the molecule thickness, which forms voids, resulting in a decreased equilibrium time [15].

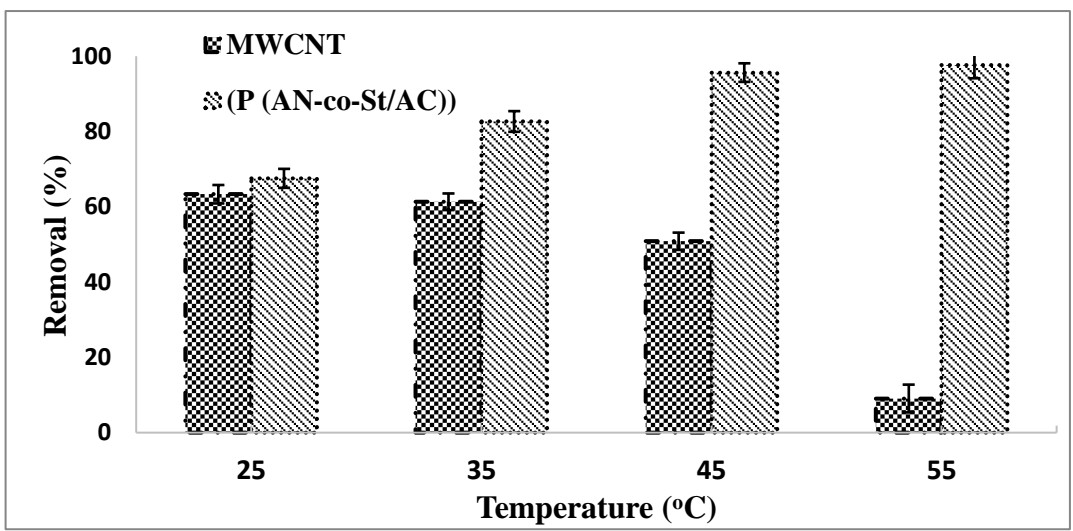

**Figure 10.** Influence of temperature on the removal of RR35.

### 3.2.6. Adsorption Thermodynamics

Thermodynamic studies were carried out to explain the effect of temperature on the adsorption. A plot of ($\Delta$Go) against (T) offers a straight line, and the values of the entropy $\Delta S°$ and enthalpy ($\Delta H°$) were calculated from slope and intercept with values (4.955 and -2.071 J mol$^{-1}$) and (392.29 and 273 J mol$^{-1}$) for MWCNTs and P (AN-co-St)/AC, respectively. The values of Gibbs free energies ($\Delta$Go) in Table 3 show that the free energy values increase with an increase in temperature shows that the adsorption technique is exothermic also it is favored by a decrease in temperature [53]. However, the great negative value of $\Delta$Go confirmed that the sorption of dye was spontaneous and feasible. In addition, the experiential data of activation energy in this work suggested that the adsorption of the RR35 onto MWCNTs and P (AN-co-St)/AC was by physisorption process, due to the range of $\Delta$Go values being between 20 and 0 kJ/mol for physisorption, while chemisorption method is between 80 and 400 kJ mol$^{-1}$ [54]. The $\Delta S°$ was found to be positive for MWCNTs. This showed that there was an increase in randomness at the solid–liquid interface, in addition to an affinity of the adsorbents for the adsorbate. While, the negative value of $\Delta S°$ for P (AN-co-St)/AC suggested a decrease in degree of freedom of the adsorbed dye [55].

**Table 3.** Thermodynamic parameters of adsorption of RR35 onto MWCNT and P (AN-co-St)/AC.

| Temperature (°C) | $\Delta G^o$ (kJ mol$^{-1}$) | | $\Delta H°$ (J mol$^{-1}$) | | $\Delta S°$ (J mol$^{-1}$) | |
|---|---|---|---|---|---|---|
| | MWCNTs | P(AN-co-St)/AC | MWCNTs | P(AN-co-St)/AC | MWCNTs | P(AN-co-St)/AC |
| 25 | −17.226 | −13.260 | | | | |
| 35 | −17.141 | −15.939 | 392.29 | 273 | 4.955 | −2.071 |
| 45 | −16.825 | −20.811 | | | | |
| 55 | −12.803 | −27.226 | | | | |

3.2.7. Regeneration Experiment

Regeneration of adsorbent composites is of economic importance. This experiment involved of numerous sorption–desorption cycles to estimate the impact of repeated decoloring on the sorption efficiency of the nanomaterials. It was found that the removal efficiency of RR35 by MWCNT and P (AN-co-St)/AC slightly changed, as shown in Figure 11. The data show that there is a difference between the original MWCNT and P (AN-co-St)/AC compared with reused nanocompounds at different initial RR35 dye concentrations. The extent of RR35 uptake reached to 98.18% after the first cycles then reduced to 62.72% after the five rounds of MWCNTs, while for P (AN-co-St)/AC the initial removal efficiency was 97.72% at 20 mg L$^{-1}$, after the first cycle, which reduced to 63.18% after the five rounds signifying its validity for regeneration as shown in Figure 11. Hence, it could be concluded that the adsorption process by MWCNTs and P (AN-co-St)/AC stays affected with extended reused cycles without losing much of its dye-adsorption capacities. As a result, two nanocompounds were utilized on a number of occasions for the removal of RR35 dye from the aqueous stage. Therefore, this suggests that they can be used more than one time, demonstrating their economic viability.

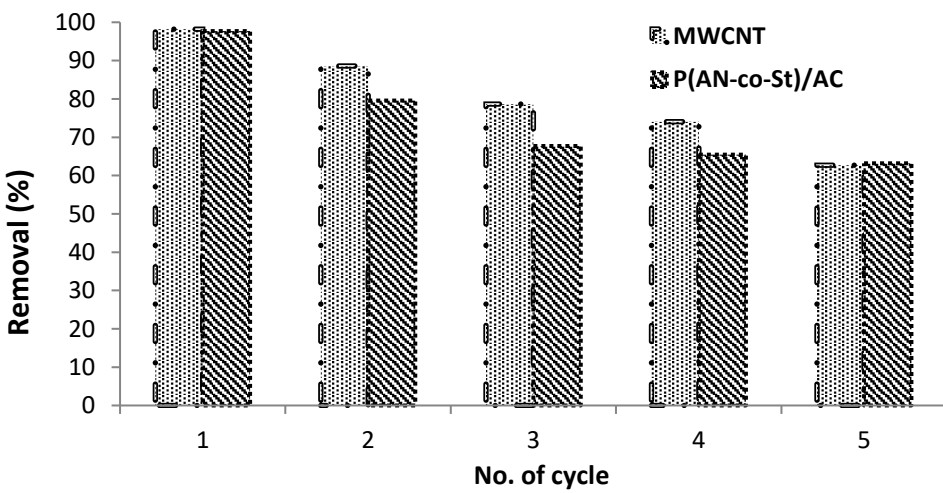

**Figure 11.** Influence of regeneration test on the removal of RR35.

*3.3. Adsorption Isotherm Study*

The sorption isotherm identified as equilibrium information is fundamental in explaining and comprehending the sorption mechanism. Four various equilibrium equations, called Langmuir, Freundlich, Tempkin, and Halsey models isotherms, were chosen to simulate the sorption isotherms and to investigate the interactions between RR35 dye and the adsorbent.

Figures 12–15 present graphs of the Langmuir, Freundlich, Tempkin, and Halsey models, respectively. The isotherm parameters obtained from these models and

correlation coefficients are shown in Table 4. The significant characteristics of the Langmuir isotherm equation, estimated as a dimensionless, steady separation factor for equilibrium factor, $R_L$, is expressed by equation in Table 1. Values of $R_L$ are defined as different types of isotherms: linear isotherm ($R_L = 1$), unfavorable isotherm ($R_L > 1$), favorable isotherm ($0 < R_L > 1$), and irreversible isotherm ($R_L = 0$) [53].

The maximum capacity $q_{max}$ (monolayer) acquired from the Langmuir equation was 256.41 and 30.30 mg g$^{-1}$ for MWCNT and P (AN-co-St)/AC, respectively. The obtained data show that the linear correlation coefficients ($R^2$) of the Freundlich, Langmuir, Tempkin, and Halsey models fit well with the investigational information for P (AN-co-St)/AC, while the models of Langmuir and Tempkin fit well with the experimental results for MWCNT. Moreover, the Freundlich and Halsey isotherm equation is not compatible with the sorption method, as the linear correlation coefficient ($R^2$) is below 0.90 for MWCNT. Favorable RR35 uptakes were established because the separation factor $R_L$ values fell within the range of from 0 to 1, with calculated values of $R_L$ of 0.008 and 0.841 for MWCNT and P (AN-co-St)/AC, respectively. Furthermore, a strong bond exists between RR35 MWCNT and P (AN-co-St)/AC adsorbents as indicated by the value of 1/n, which is named the heterogeneity parameter, describing the deviation from the linearity of sorption as follows: where 1/n is equivalent to 1, the adsorption is linear, and the dye particle concentration does not influence the division between the two stages. When 1/n is below 1, chemical adsorption occurs, and this demonstrates an ordinary Langmuir isotherm; however, when 1/n is more than 1, cooperative adsorption occurs, and this adsorption is more favorable physically and includes strong interactions among the particles of the adsorbate [54]. In the present study, the values of parameter "1/n" are less than 1; the data show a chemical adsorption process on a surface with this model to be favorable.

**Table 4.** Isotherm parameter of RR35 adsorption onto MWCNT and P(AN-co-St)/AC nanocomposites.

| Isotherm model | Parameters | MWCNT | P(AN-co-St)/AC |
|---|---|---|---|
| Langmuir | $R^2$ | 0.987 | 0.998 |
| | $q_{max}$ (mg g$^{-1}$) | 256.41 | 30.30 |
| | b | 0.29 | 0.0063 |
| | $R_L$ | 0.008 | 0.814 |
| Freundlich | $R^2$ | 0.732 | 0.996 |
| | 1/n | 0.56 | 0.725 |
| | $K_f$ | 5.57 | 24.02 |
| Tempkin | $R^2$ | 0.928 | 0.953 |
| | A | 64.06 | 2.10 |
| | B | 13.36 | 32.05 |
| | $b_T$ | 185.44 | 77.29 |
| Halsey | $R^2$ | 0.876 | 0.996 |
| | $1/n_H$ | 0.3006 | 0.725 |
| | $K_H$ | 82.51 | 34.21 |

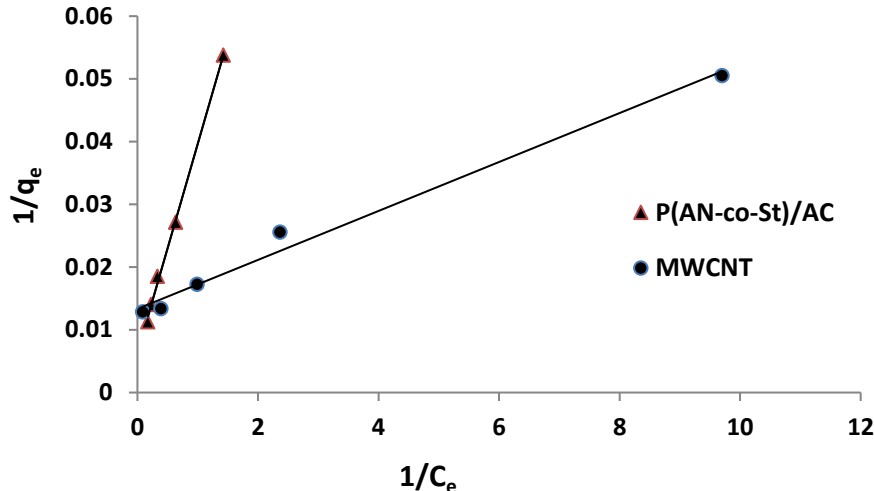

**Figure 12.** Langmuir plots for adsorption of RR35.

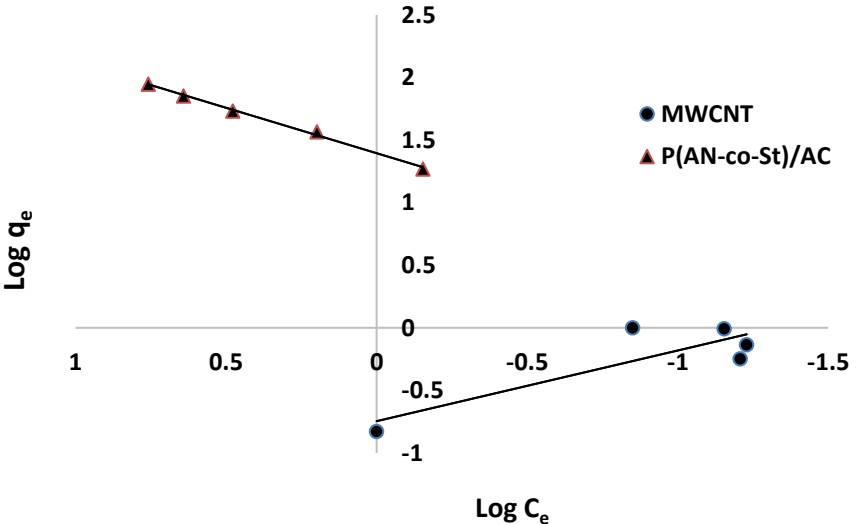

**Figure 13.** Freundlich plots for adsorption of RR35.

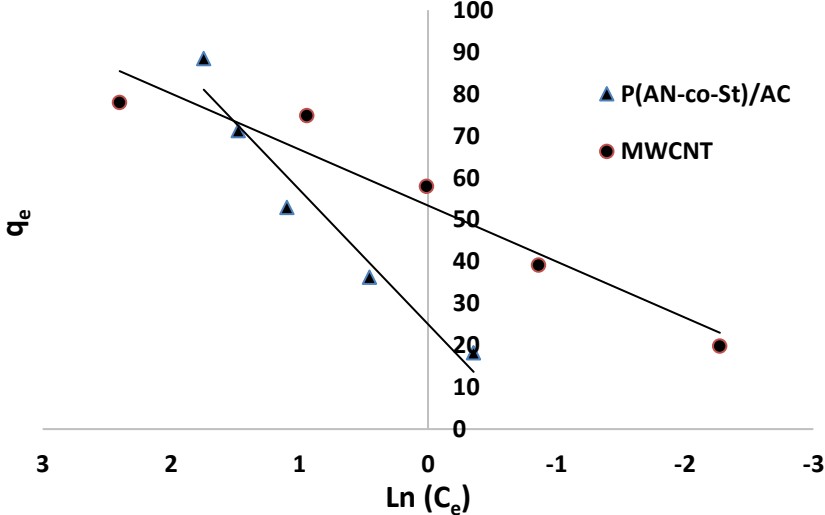

**Figure 14.** Tempkin plots for adsorption of RR35.

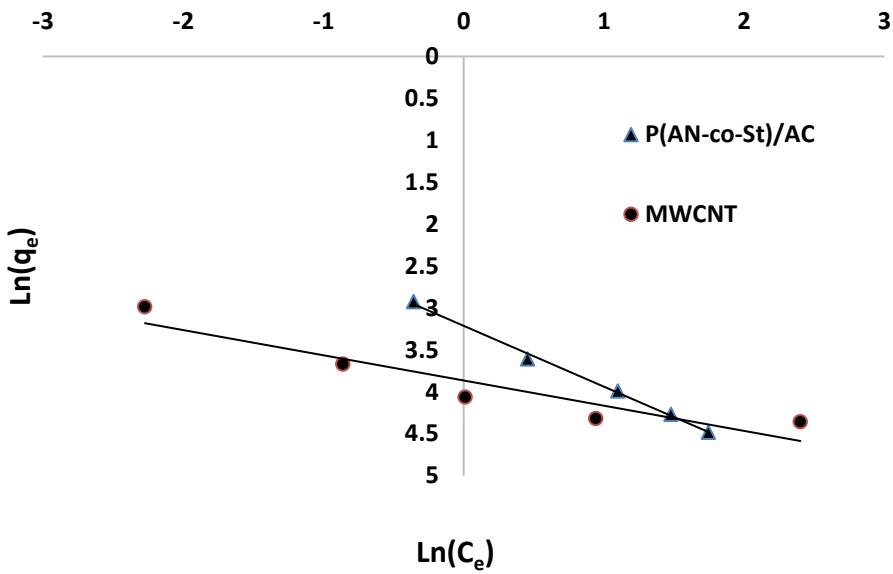

**Figure 15.** Halsey plots for the removal of RR35 onto different compounds.

### 3.4. Kinetic Studies

Three kinetic equations, namely, pseudo first order, pseudo second order, and intraparticle diffusion, were employed to determine the nature and rate of RR35 dye removal by MWCNTs and P (AN-co-St)/AC. The basic linearized form of the three kinetic models was addressed by the equations as presented in Table 5.

**Table 5.** The integral linearized form of kinetic models.

| Models | Equation | Definition | Ref |
|---|---|---|---|
| Pseudo first order (PFO) | $Log\ (q_e-q_t) = log\ q_e - (k_1/2.303)t$ | $q_e$, $q_t$ are, respectively, represent adsorption capacity at equilibrium and at time t (mg $g^{-1}$). $k_1$: pseudo first-order rate constant ($min^{-1}$) | [55] |
| Pseudo second order (PSO) | $t/q_t = 1/K_2q_e^2 + (1/q_e)\ t$ | $k_2$: the pseudo second-order rate constant (g $mg^{-1}$ min) | [56] |
| Intraparticle diffusion (IPD) | $q_t = K_{dif}\ t^{1/2} + C$ | $C$: the intercept in addition; $K_{dif}$: the intraparticle diffusion rate constant (mg $g^{-1}$ $min^{1/2}$) | [57] |

Table 6 exhibits the fitting data for the three kinetic equations. The determined adsorption capacity $q_e$ (cal) from the PFO equation differs fundamentally from investigational values ($q_e$ exp). The linear correlation coefficients and $R^2$ values were not shut to 1; this infers that the PFO equation was not appropriate to expose the adsorption mechanism of RR35 to the composite adsorbents. On the other hand, from Table 7, it can be seen that the PSO kinetic equations fitted considerably well with the investigational data. The correlation coefficient values are near unity, demonstrating good agreement. In addition, the predicted ($q_e$ cal) values are near the corresponding experimental values ($q_e$ exp). Hence, the PSO equation fit well with the RR35 adsorption process by MWCNTs and P(AN-co-AS)/AC (Figures 16 and 17). The intraparticle diffusion model provided low fit according to $R^2$ values, for P (AN-co-AS)/AC and MWCNTs [58].

**Table 6.** Kinetic parameters of RR35 adsorption onto MWCNT and P(AN-co-AS)/AC nanocomposites.

| Model | Parameters | Values | |
|---|---|---|---|
| | | MWCNT | P (AN-co-ST)/AC |
| First-order kinetic | $q_e$ (exp.) (mg g$^{-1}$) | 33.64 | 31 |
| | $q_e$ (calc.) (mg g$^{-1}$) | 61.51 | 39.61 |
| | $K_1$ (1/min) | 0.043 | 0.0651 |
| | $R^2$ | 0.167 | 0.666 |
| Pseudo second-order kinetic | $q_e$ (exp.) (mg g$^{-1}$) | 33.64 | 31 |
| | $q_e$ (calc.) (mg g$^{-1}$) | 25.51 | 21.36 |
| | $K_2$ (mg g$^{-1}$ min$^{-1}$) | 0.054 | 0.047 |
| | $R^2$ | 0.998 | 0.905 |
| Intraparticle diffusion | K | −0.032 | 0.688 |
| | C | 0.791 | 18.698 |
| | $R^2$ | 0.178 | 0.066 |

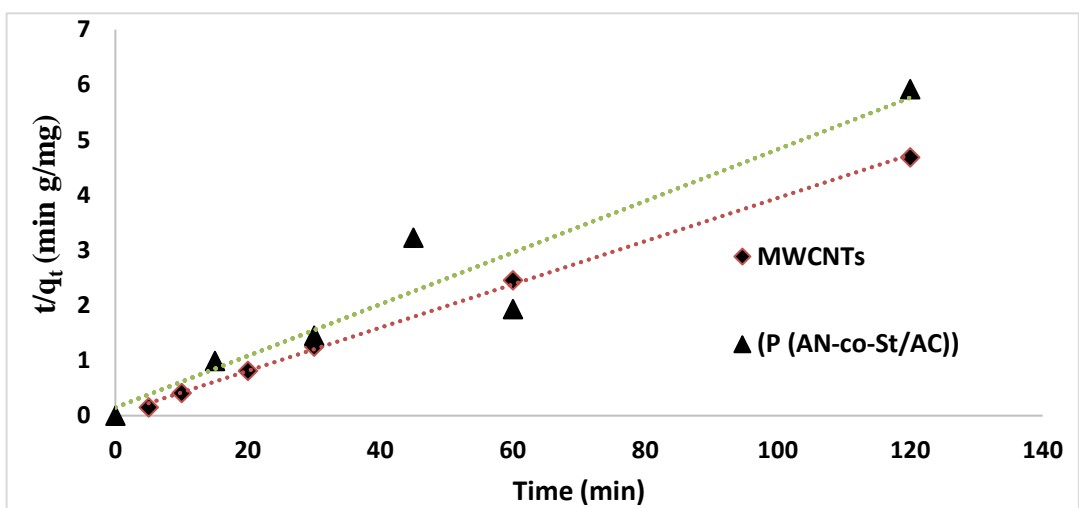

**Figure 16.** Second-order kinetic for the removal of RR35.

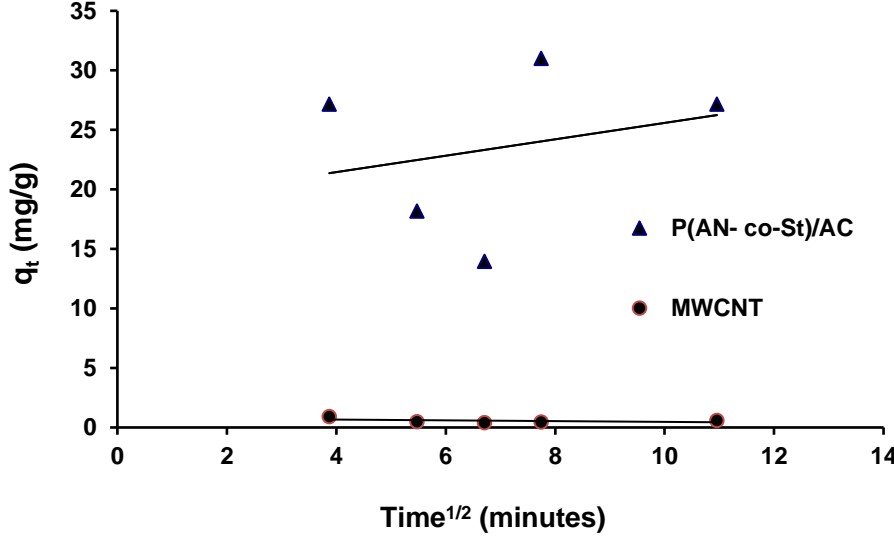

**Figure 17.** Intraparticle diffusion kinetic for the removal of RR35.

### 3.5. Comparison of MWCNT with Other Adsorbents

Comparison of the higher RR35 adsorption capacities of MWCNT adsorbent with that of the other previously study adsorbents was performed and is exposed in Table 7, which reflects probable efficacy for MWCNT and P (AN-co-AS)/AC utilization in RR35 removal.

**Table 7.** Comparison of the removal of dyes from wastewater solutions by P(AN-co-St) and various MWCNT.

| Adsorbent | Dye | Adsorption capacity $(mg \, g^{-1})$ | Ref. |
|---|---|---|---|
| Oxidize MWCNTs | Bromothymol blue | 55 | [59] |
| MWCNTs | methylene blue | 64.7 | [60] |
| MWCNTs | Methyl orange | 52.86 | [61] |
| MWNTs | Congo red dye | 352.11 | [17] |
| Carbon nanotubes | methyl orange | 78.07 | [15] |
| poly acrylonitrile | Methylene blue | 8.7600 | [62] |
| P (styrene-co-acrylonitrile) | Methylene blue | 17.77 | [19] |
| Poly (AN-co-VP)/Zeolite | Brilliant green | 23.81 | [20] |
| P (AN-co-St)/AC | Reactive red 35 | 256.41 | This study |
| MWCNTs | Reactive red 35 | 30.30 | This study |

## 4. Conclusions

This study demonstrated that the purified MWCNTs and poly (acrylonitrile-co-styrene)/AC in this work are effective adsorbents in complete elimination of RR35 dye from aqueous solution. MWCNTs were prepared by the catalytic chemical vapor deposition (CCVD) method, a cheap technique, and the poly (acrylonitrile-co-styrene)/AC composite was prepared by a simple precipitation polymerization process. The structures of MWCNTs and P (AN-co-ST)/AC were confirmed by particle size analysis, Fourier transform infrared, and scanning electron microscope, and the X-ray diffraction analysis technique was applied to investigate the crystal structure of MWCNTs. According to FTIR examination, the main functional groups involved in the adsorption of RR35 are C=O, N–H, thiol S–H, C–H–$NH_2$, O–H, C=C, and C≡N. The batch process conditions have a significant influence on the removal process, where increasing the adsorbent dose has a directly proportional effect on the RR35 dye removal process. In addition, it was shown that the adsorption of RR35 is favored at a low dye concentration and pH. Furthermore, thermodynamic factors of the adsorption process ($\Delta G°$, $\Delta H°$, and $\Delta S°$) were determined. The adsorption of IV2R dye was exothermic, a spontaneous process, and the reaction of adsorption is a physisorption process. Adsorption isotherm data were described by Langmuir, Freundlich, Tempkin, and Halsey isotherm models and the equilibrium data for the MWCNT adsorbent fit well in the Langmuir isotherm model, with higher adsorption capacities of 256.41 and 30.3 mg $g^{-1}$ for MWCNTs and P (AN-co-St)/AC, respectively. The kinetic data display a poor pseudo first-order fit to the experimental results. Conversely, the pseudo second order and intraparticle diffusion show a good fit to the experimental data. Conventional treatment technologies for elimination of dyes from aqueous effluents for example ozonation, membrane separation, flocculation, coagulation, and aerobic or anaerobic process are not economical, expensive, and difficult to operate and produce huge amounts of toxic chemical sludge. Nonetheless, in this technique application, the low cost of the prepared materials, the fast adsorption phenomenon occur at the shorter contact time with small amounts of adsorbents at room temperature and its regeneration for more cycles with high efficiency, which allows for a smaller size of the contact apparatus, which in turn directly affects both the operation cost and capacity of the technique.

Therefore, this research confirms that adsorption of RR35 dye by using MWCNTs and the combination of poly (AN-co-St) with AC nanocompounds led to an efficient improvement in the removal process of dye from aqueous solution in a short adsorption time.

**Author Contributions:** Conceptualization, A.E.A.; methodology, A.E.A., K.M.A., and M.A.A.-S.; software, A.E.A. and M.A.A.-S.; formal analysis, A.E.A.; investigation, A.E.A., K.M.A., and M.A.A.-S.; resources, A.E.A. and K.M.A.; data curation, A.E.A., M.A., A.T.M., and M.A.A.-S.; writing—original draft preparation, A.E.A., M.A., and A.T.M.; writing—review and editing, A.E.A., M.A., and A.T.M.; visualization, K.M.A. and M.A.; supervision, K.M.A. and M.A.A.-S.; project administration, A.E.A.; funding acquisition, K.M.A. All authors have read and agreed to the published version of the manuscript.

**Funding:** This research was funded by Taif University Researchers Supporting Project number TURSP-2020/267, Taif University, Taif, Saudi Arabia.

**Institutional Review Board Statement:** Not applicable

**Informed Consent Statement:** Not applicable

**Data Availability Statement:** All relevant data are within the paper, and those are available at the corresponding author.

**Acknowledgments:** The authors appreciated Taif University Researchers Supporting Project number TURSP-2020/267, Taif University, Taif, Saudi Arabia.

**Conflicts of Interest:** The authors declare no conflict of interest.

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
