# Peer review of "Removing of Anionic Dye from Aqueous Solutions by Adsorption Using of Multiwalled Carbon Nanotubes and Poly (Acrylonitrile-styrene) Impregnated with Activated Carbon"

_sustainability, doi:10.3390/su13137077_

Round 1
Reviewer 1 Report
Overall Review: Abulnaja et al. have synthesized MWCNTs and an activated carbon/P(AN-co-ST) nanocomposite, and aimed to determine the sorption capacities. While the authors were able to show the sorptive capacities of each nanomaterial, this work has many issues. The comparison of MWCNT and an AC/polymer nanocomposite were very strange and seemingly unrelated. MWCNT adsorption of nonionic dyes has been established in the literature for over a decade, and this paper unfortunately does not add any new information to that literature. As for the AC/polymer nanocomposite, no work was done to study the AC, alone, or the polymer, alone, as a sorbent. Asa result, the mechanism of action is unclear, and it is entirely possible that the addition of polymer to the surface of AC might have actually acted to reduce the adsorption capacity of just AC. This paper is likely not novel enough or robust enough to be published in Sustainability. More detailed responses can be found below.
Detailed Review:
- This paper unfortunately feels quite disconnected, and seems to be answering two completely different questions. The first question is getting at the sorptive capacity of MWCNTS, while the other aims to determine the activate carbon/polymer nano composite sorptive capacity. These two nanomaterials are significantly different, and there is no real reason to write about them both in the same article, especially since the conclusions were not robust for either of the two samples studied.
- The sorption capacity of MWCNTs, especially those that have been functionalized by acid treatment, is very well established in the literature. This current study only considers one dye, and only considered one sample of MWCNTs, while others in the literature over the past decade (https://doi.org/10.1002/slct.201700135, https://doi.org/10.1021/je3001552, https://doi.org/10.1016/j.cis.2013.03.003, https://doi.org/10.1039/C7RA09377B) study the removal of a variety of dyes using MWCNTs with multiple physical and chemical modifications. As a result, each of these studies is able to better determine a physical or chemical mechanism for adsorption based on varied MWCNT properties. This paper, as it currently stands, is not nearly as robust, and its findings with regards to MWCNT do not add much to the present literature.
- It has been well-established in the literature that the “size” of MWCNTs in solution, especially with oxygen functionalities, does not always match up with the size of the dry particles (https://doi.org/10.1016/j.carbon.2019.08.063). Further, it is also noted (in the same article link) that surface functionalization does not necessarily make the MWCNT stable in solution. As such, dynamic light scattering should be included to determine the hydrodynamic radius of MWCNT (and the AC/polymer composite) in solution, while a time-resolved DLS study should be used to ensure the nanomaterials are stable in aqueous solution. There is no proof in the current work that the composites are stable.
- This manuscript did not properly address the mechanism by which the polymer/AC nanocomposite works. There was no work showing the sorption behavior of the AC, alone. There is also no work showing the sorption behavior of the polymer, alone. For all we know, the AC might have performed better without the polymer than with the polymer, since they polymer could potentially be sterically hindering sorptive sites on the AC. As a result, the authors must explicitly study just AS, just the polymer, and the polymer/AC nanocomposite (at the bare minimum).
- Almost no characterization of the AC was done at all. It is incredibly important to know that there are hundreds of varieties of activated carbon, many of which have extremely different properties from one another. A lack of characterization of AC does the reader no good, and weakens the paper greatly.
- Figure 2 was not needed and offers no additional value to the paper.
- Table 2 is just a repetition of section 3.1.1 and is likely unneeded.
- Figure 3: (a) and (b) are reversed. Further, based on the spectra of (a), it is unclear if there actually are any peaks at the locations indicated. I would suggest redoing the FTIR, or supplementing with another technique like XPS.
- Table 3: Mean particle size was undefined (was it radius, diameter, etc.?) It is a very strange measure to use here, since MWCNTs are closer to 1-dimensional, while the AC/polymer seemed to be more amorphous/3-dimensional. MWCNTs are therefore very long, which is not captured by a simple average size measurement. As a result, this measurement is likely not appropriate and doesn’t effectively compare these materials at all.
- Figure 4: The image quality here needs to be improved. It is difficult to draw any conclusions from the current images.
- Raman spectroscopy: if Raman is ever performed on carbon nanotubes or other graphitic materials, it is essential that the D:G ratio is reported, since that holds a lot of information inside of it.
- This is a study on sorbents, and sorption capacity is partially a function of surface area. As such, it is crucial that the specific surface area of each sorbent be explicitly reported. This is a huge omission that must be corrected.
- While each of nanomaterials was characterized in section 3.1, none of that information was used throughout the rest of the study, and most of these characterizations were somewhat superficial. More appropriate characterizations should be chosen, they should relate strongly to the results and conclusions being drawn, and those relationships should be explicitly noted by the authors. Some suggestions include DLS, TR-DLS, XPS, BET, etc.
- Section 3.2.1 – pH changes can strongly impact the stability of MWCNT in solution (https://doi.org/10.1021/es801251c). MWCNT aggregates do not have the same sorption capacity as primary particles do, so I again note that a TR-DLS technique should be used to ensure particle stability at various pH values. Otherwise, your results may be a function of aggregation, and not of the material itself. Note that the same concern exists for the polymer/AC nanocomposite.
- Section 3.2.5 – the authors made no true attempt at explaining why removal rate for MWCNTs went down with increasing temperature (against common knowledge). An experiment is needed to explain why these differences occur. This could include surface energy calculations, or something similar, but right now, that conclusion is not robust at all.
- Section 3.2.6- this regeneration study is very underwhelming. To prove regeneration potential, I would suggest at least 5-8 cycles (at the very minimum) to show the impacts of regeneration over time. One use-regeneration-reuse cycle is wholly insufficient to draw any real conclusions.
- Equation (3)- this seems like it belongs in Table 1 instead.
- While imperfect English in manuscripts is not too big of an issue, the grammatical errors in this manuscript tended to be distracting. I would suggest including a native English speaker as an editor for the next draft.
Author Response
SUMMARY OF AUTHOR(S) RESPONSE TO REVIEWER’S COMMENTS
Manuscript ID: sustainability 1212297
Title: Removing of Anionic Dye from Aqueous Solutions by Adsorption Using of Multiwalled Carbon Nanotubes and Poly (Acrylonitrile-Styrene) Impregnated With Activated Carbon.
Authors: Khamael M. Abualnaja, Ahmed E. Alprol, M.A. Abu-Saied, Mohamed Ashour, Abdallah Tageldein Mansour
Reviewer 1# round 1 Comment |
Author(s) response |
Major Comments: |
|
Overall Review: Abualnaja et al. have synthesized MWCNTs and an activated carbon/P(AN-co-ST) nanocomposite, and aimed to determine the sorption capacities. While the authors were able to show the sportive capacities of each nanomaterial, this work has many issues. The comparison of MWCNT and an AC/polymer nanocomposite were very strange and seemingly unrelated. MWCNT adsorption of nonionic dyes has been established in the literature for over a decade, and this paper unfortunately does not add any new information to that literature. As for the AC/polymer nanocomposite, no work was done to study the AC, alone, or the polymer, alone, as a sorbent. As a result, the mechanism of action is unclear, and it is entirely possible that the addition of polymer to the surface of AC might have actually acted to reduce the adsorption capacity of just AC. This paper is likely not novel enough or robust enough to be published in Sustainability. More detailed responses can be found below. |
|
This paper unfortunately feels quite disconnected, and seems to be answering two completely different questions. The first question is getting at the sportive capacity of MWCNTS, while the other aims to determine the activate carbon/polymer nano composite sportive capacity. These two nanomaterials are significantly different, and there is no real reason to write about them both in the same article, especially since the conclusions were not robust for either of the two samples studied.
|
- This manuscript aims to investigate the effect of MWCNT ability to remove Reactive Red 35 dye from aqueous solutions, on the other hand, to study the ability of activated carbon mixed with poly (acrylonitrile-co-styrene) to remove the same dye. The common factor (pollutant) here is the same dye, as we study two different materials to remove the same pollutant. -In comparison, the results showed that the amount of adsorbed dye was increased from from 18.30 mg g-1 to 78.73 mg g-1 with different conc of MWCNT, while with different conc of P(AN-co-St)/AC, rises from 18.26 mg g-1 to 88.75 mg g-1 by increasing the initial dye concentration from 10 mg L-1 to 50 mg L-1. - Thus, we hope our explains are sufficient for these important two questions. |
The sorption capacity of MWCNTs, especially those that have been functionalized by acid treatment, is very well established in the literature. This current study only considers one dye, and only considered one sample of MWCNTs, while others in the literature over the past decade (https://doi.org/10.1002/slct.201700135, https://doi.org/10.1021/je3001552, https://doi.org/10.1016/j.cis.2013.03.003, https://doi.org/10.1039/C7RA09377B) study the removal of a variety of dyes using MWCNTs with multiple physical and chemical modifications. As a result, each of these studies is able to better determine a physical or chemical mechanism for adsorption based on varied MWCNT properties. This paper, as it currently stands, is not nearly as robust, and its findings with regards to MWCNT do not add much to the present literature. |
- The main objective of this study was to prepare and test the behavior of the different materials (P(AN-co-St)/AC, and MWCNTs) with the experimented environmental conditions. -In addition this study demonstrated that the (P (AN-co-St)/AC, and MWCNTs) had an excellent adsorption performance, in comparison with many studies have been conducted to remove one dye from aqueous solution by “one material” and/or “two different materials – such as the same idea of current manuscript”, as mentioned in the following literature: - for MWCNT http://dx.doi.org/doi:10.1039/C2CP41475A http://dx.doi.org/10.1016/j.molliq.2015.09.027 http://dx.doi.org/10.1016/j.proenv.2013.04.120; http://dx.doi.org/10.1016/j.cej.2012.08.025 httpI://dx.doi.org/10.1590/S1516-14392013005000204 http://dx.doi.org/10.1155/2014/201052; https://doi.org/10.1186/s40068-020-00191-4). - for poly (acrylonitrile-co-styrene): https://doi.org/10.1007/s13762-018-02199-x; http://dx.doi.org/10.1080/19443994.2012.691688; http://dx.doi.org/10.1080/19443994.2015.1080192 http://dx.doi.org/10.5004/dwt.2020.24967 http://dx.doi.org/doi:10.5004/dwt.2020.24911 https://doi.org/10.1016/j.molliq.2019.111335 |
It has been well-established in the literature that the “size” of MWCNTs in solution, especially with oxygen functionalities, does not always match up with the size of the dry particles (https://doi.org/10.1016/j.carbon.2019.08.063). Further, it is also noted (in the same article link) that surface functionalization does not necessarily make the MWCNT stable in solution. As such, dynamic light scattering should be included to determine the hydrodynamic radius of MWCNT (and the AC/polymer composite) in solution, while a time-resolved DLS study should be used to ensure the nanomaterials are stable in aqueous solution. There is no proof in the current work that the composites are stable. |
- The DLS technique is not available. -A particle size was investigated by particle size analyzer (Beckman Coulter, USA) which its measure nature distribution of the substances in the water, in this study of removing the dye occurs in aqueous solutions, therefore the volume was stable, and previous studies confirmed that the size of the particles in the wet samples is more stable. - As for the following study (https://doi.org/10.1016/j.carbon.2019.08.063), which has been studied the behavior of the MWCNT-induced toxicity, by understanding the role of aggregation and the chorion in embryonic zebra fish studies. The estimated primary particle or agglomerate size was plotted over time to evaluate aggregation kinetics of the O-MWCNTs in stock solution during the first 3 h of exposure to zebrafish embryos. |
This manuscript did not properly address the mechanism by which the polymer/AC nanocomposite works. There was no work showing the sorption behavior of the AC, alone. There is also no work showing the sorption behavior of the polymer, alone. For all we know, the AC might have performed better without the polymer than with the polymer, since they polymer could potentially be sterically hindering sorptive sites on the AC. As a result, the authors must explicitly study just AS, just the polymer, and the polymer/AC nanocomposite (at the bare minimum). |
- Characterization of Poly (Acrylonitrile-Styrene) were added at Figures (2C, 3A & 4A) and Table 2. - On the other hand, this material “Poly(Acrylonitrile-Styrene” has been used to remove dyes from arouses solution, as well as, it was previously used (https://doi.org/10.3390/nano11051144), and resulted good findings in the removing of dye from aqueous solutions. In this study when adding to activated carbon to this material (Poly (Acrylonitrile-Styrene)) gave better results.
|
Almost no characterization of the AC was done at all. It is incredibly important to know that there are hundreds of varieties of activated carbon, many of which have extremely different properties from one another. A lack of characterization of AC does the reader no good, and weakens the paper greatly.
|
- At this point, we relied on the characterization of the activated carbon by knowing the properties of the polymer that includes the active carbon and the polymer without the active carbon, and thus we were able to know the changes that resulted from the presence of the active carbon. - Where the polymer without activated carbon properties have been added (SEM, FTIR, Raman, PSA), Figs (2C, 3A, 4A) and Table 2. |
Figure 2 was not needed and offers no additional value to the paper. |
- Authors agree with this valuable comment. This Figure was deleted. |
Table 2 is just a repetition of section 3.1.1 and is likely unneeded. |
- This Table was deleted. |
Figure 3: (a) and (b) are reversed. Further, based on the spectra of (a), it is unclear if there actually are any peaks at the locations indicated. I would suggest redoing the FTIR, or supplementing with another technique like XPS. |
- Authors agree with this valuable suggest. - Figure 2 has been redrawn (Page 6). -Figure 3 (FTIR) replaced to Figure 2, because Figure 2 was deleted. |
Table 3: Mean particle size was undefined (was it radius, diameter, etc.?) It is a very strange measure to use here, since MWCNTs are closer to 1-dimensional, while the AC/polymer seemed to be more amorphous/3-dimensional. MWCNTs are therefore very long, which is not captured by a simple average size measurement. As a result, this measurement is likely not appropriate and doesn’t effectively compare these materials at all. |
- Mean particle size was defined already in the manuscript, kindly please revise the following: - (Page: 7, Lines: 242 – 243), and - Table 2 (row 3).
|
Figure 4: The image quality here needs to be improved. It is difficult to draw any conclusions from the current image |
-Figure 3 has been redrawn. - Figure 4 has been renumbered to be Figure 3 |
Raman spectroscopy: if Raman is ever performed on carbon nanotubes or other graphitic materials, it is essential that the D: G ratio is reported, since that holds a lot of information inside of it. |
-D: G ratio is reported in manuscript (Page: 8 and 9, Lines: 264 – 279). |
This is a study on sorbents, and sorption capacity is partially a function of surface area. As such, it is crucial that the specific surface area of each sorbent be explicitly reported. This is a huge omission that must be corrected.
|
-The results showed that the amount of dye adsorbed per unit mass of adsorbent increased from 18.30 mg g-1 to 78.73 mg g-1 for MWCNT, while for P(AN-co-St)/AC, rises from 18.26 mg g-1 to 88.75 mg g-1 by increasing the initial dye concentration from 10 mg L-1 to 50 mg L-1. (Page: 12, Lines: 351-356). - These findings were agreed with the results reported by other studies [Please, find Ref. 28, 39, 41, 42, and 43]. |
While each of nanomaterials was characterized in section 3.1, none of that information was used throughout the rest of the study, and most of these characterizations were somewhat superficial. More appropriate characterizations should be chosen, they should relate strongly to the results and conclusions being drawn, and those relationships should be explicitly noted by the authors. Some suggestions include DLS, TR-DLS, XPS, BET, etc.
|
- Authors agree with this valuable suggest and believe that this suggest added important value to our manuscript. Please, find the further results, discussions, and references which have been added as the following: - Figures (2C, 3A, and 4A) - Page 4, Lines: 171-173 - Page: 5, Lines: 203-204 - Page: 6, Lines: 206-208. - Page: 7, Lines: 222-238; 240- 244; 247-248; and 252-254. - Page: 8 and 9, Lines: 264-279. - Page 10, Lines: 291-300 (3.1.6. BET Analysis was added) About: DLS, TR-DLS, and XPS, actually, it is not available to us, while in the future work we will do. |
Section 3.2.1 – pH changes can strongly impact the stability of MWCNT in solution (https://doi.org/10.1021/es801251c). MWCNT aggregates do not have the same sorption capacity as primary particles do, so I again note that a TR-DLS technique should be used to ensure particle stability at various pH values. Otherwise, your results may be a function of aggregation, and not of the material itself. Note that the same concern exists for the polymer/AC nanocomposite. |
-The authors agree with this valuable suggestion. While unfortunately, the amount of available material is not sufficient to do such an analysis. Moreover, it is difficult to do this analysis because of the Covid-19 coronavirus pandemic. While this analysis will be considered in the next work.
|
Section 3.2.5 – the authors made no true attempt at explaining why removal rate for MWCNTs went down with increasing temperature (against common knowledge). An experiment is needed to explain why these differences occur. This could include surface energy calculations, or something similar, but right now, that conclusion is not robust at all. |
- Authors agree with this valuable suggest and believe that this suggest added important value to our manuscript. - To explain differences between increasing temperature and removal rate for both adsorbents, the “3.2.6. Adsorption thermodynamics” have been added (Page: 14 and 15, Lines: 409-425 and Table 3)
|
Section 3.2.6- this regeneration study is very underwhelming. To prove regeneration potential, I would suggest at least 5-8 cycles (at the very minimum) to show the impacts of regeneration over time. One use-regeneration-reuse cycle is wholly insufficient to draw any real conclusions. |
Actually, MWCNTs and P(AN-co-St)/AC may be reused for four times due to its economic probability, -MWCNTs and P(AN-co-St)/AC signified validity for reuse after 5 cycles of regenerations as showed in manuscript (Page: 15, Lines: 429-431; and 435-441). |
Equation (3) - this seems like it belongs in Table 1 instead. |
- Done |
While imperfect English in manuscripts is not too big of an issue, the grammatical errors in this manuscript tended to be distracting. I would suggest including a native English speaker as an editor for the next draft. |
- The manuscript has been check by native English Speaker from authorized center of MDPI (the certificate attached as supplementary file).
|
We would like to extend our sincere thanks and appreciation to the anonymous reviewer #1. In fact, the reviewer comments, suggests, and guidance added important value to the manuscript and increased its scientific content. Therefore, the words cannot express their gratitude for their time and effort they put in evaluating this research.

Reviewer 2 Report
- My first concern is that the contribution of the paper is not clearly described in the manuscript yet. The authors wrote the aim of the paper is to remove the dye using MWCNTs and P(AN-co-ST). Like the authors cited in the intro, both materials have been shown its capability to remove dyes in other works.
- You could make an insert figure for FTIR of MWCNT in the range of 3000 – 3800 cm-1.
- There’re many typos and description errors. For example, 9.51 e13 is the coefficient of polymer composite not MWCNTs and the -63.915 comes from nowhere. (row230)
- In Figure 4 (D), there’s no scale bar. The green labels in Figure 4 (B) are quite blurring.
- Can you enlarge figures like Ramen proportionally? And also format all the figures. Currently they are in different sizes. That will make them look much better.
Author Response
SUMMARY OF AUTHOR(S) RESPONSE TO REVIEWER’S COMMENTS
Manuscript ID: sustainability 1212297
Title: Removing of Anionic Dye from Aqueous Solutions by Adsorption Using of Multiwalled Carbon Nanotubes and Poly (Acrylonitrile-Styrene) Impregnated With Activated Carbon.
Authors: Khamael M. Abualnaja, Ahmed E. Alprol, M.A. Abu-Saied, Mohamed Ashour, Abdallah Tageldein Mansour
Reviewer 1# round 1 Comment |
Author(s) response |
Comments: |
|
My first concern is that the contribution of the paper is not clearly described in the manuscript yet. The authors wrote the aim of the paper is to remove the dye using MWCNTs and P(AN-co-ST). Like the authors cited in the intro, both materials have been shown its capability to remove dyes in other works. |
|
You could make an insert figure for FTIR of MWCNT in the range of 3000 – 3800 cm-1.
|
- Thank you for this valuable comment. FTIR of MWCNT was inserting in Figure 2, at Page 6, also, please find (Page: 6, Lines: 206-208) and (Page: 7, Lines: 214-215). |
There’re many typos and description errors. For example, 9.51 e13 is the coefficient of polymer composite not MWCNTs and the -63.915 comes from nowhere. (row230) |
- The diffusion coefficient, Counts/s and Baseline error was delegated from table 2 and text. - Page 7. |
In Figure 4 (D), there’s no scale bar. The green labels in Figure 4 (B) are quite blurring.
|
- Note: Figure 4 replaced to Figure 3, because Figure 2 was deleted according to the request of reviewer 1# - Figure 3 has been redrawn and scale bar has been added (Page 8). |
Can you enlarge figures like Ramen proportionally? And also format all the figures. Currently, they are in different sizes. That will make them look much better. |
- Thank you for this valuable comment. All figures have been redrawn to be clearer. - Moreover, the manuscript has been check by a native English Speaker from the authorized center of MDPI (the certificate attached as a supplementary file). |
We would like to extend our sincere thanks and appreciation to anonymous reviewer #2. In fact, the reviewer comments, suggestions, and guidance added important value to the manuscript and increased its scientific content. Therefore, the words cannot express their gratitude for their time and effort they put into evaluating this research.

Reviewer 3 Report
The manuscript presented the efficiency of using hybrid component-material techniques to treat dye contaminants. Although this study tries to cover a wide range of variable effective on separation efficiency and component fabrication and characterization, there are a few problems (grammatical and methodological) needed to be addressed:
- The English writing of the manuscript is not in the level of academic journals and needs to be restructured, using active voice, correct and accurate terms and wording.
- Abstract needs to be rewritten, the results are not presented well.
- In conclusion section where the authors discussed about economical advantage of current technique, please provide a brief economical comparison with the other related techniques.
- the unites are not same and should be in a consistent throughout the article (e.g. mg/l or mg/L?)
- Figures and equations should be revised by the authors (e.g., Fig1 and equation 1, 2) and Fig 2 is not clear
- Figure 3 , 4, 5, 6, and 17 are in different styles and colors and do not have units ( e.g. wave length)
- For better understanding of nano tube separation, this section needs to include the analysis of the surface Zeta potential under different conditions (pH or concentrations)
- All the references need to be revised (e.g. 1,14, 16, 18, many more) and some need to be updated (21, 24 are too old)
- Reference 27 and 33 are repetitive
Author Response
SUMMARY OF AUTHOR(S) RESPONSE TO REVIEWER’S COMMENTS
Manuscript ID: sustainability 1212297
Title: Removing of Anionic Dye from Aqueous Solutions by Adsorption Using of Multiwalled Carbon Nanotubes and Poly (Acrylonitrile-Styrene) Impregnated With Activated Carbon.
Authors: Khamael M. Abualnaja, Ahmed E. Alprol, M.A. Abu-Saied, Mohamed Ashour, Abdallah Tageldein Mansour
Reviewer 3# R1 Comment |
Author(s) response |
Comments: |
|
The manuscript presented the efficiency of using hybrid component-material techniques to treat dye contaminants. Although this study tries to cover a wide range of variable effective on separation efficiency and component fabrication and characterization, there are a few problems (grammatical and methodological) needed to be addressed:
|
|
The English writing of the manuscript is not in the level of academic journals and needs to be restructured, using active voice, correct and accurate terms and wording. |
- The manuscript has been check by a native English Speaker from authorized center of MDPI (the certificate attached as supplementary file).
|
Abstract needs to be rewritten, the results are not presented well. |
- The Abstract has been improved and supplemented with the principal results (Page: 1, Lines: 25 - 35). |
In conclusion section where the authors discussed about economic advantage of current technique, please provide a brief economical comparison with the other related techniques. |
-The conclusion section has been rewritten, improved, and provided with the obtained results (Page: 20, Lines: 532-533, 535-540). Moreover, the economic advantage of current technique has been added (Page: 20, Lines: 545-556). |
The unites are not same and should be in a consistent throughout the article (e.g. mg/l or mg/L?) |
- The units have been consistent throughout the article (mg L-1), according to the journal requirement. Thank you for your observation. |
Figures and equations should be revised by the authors (e.g., Fig1 and equation 1, 2) and Fig 2 is not clear |
- All Figure and Equations have been revised. - Note: Figure 2 was deleted according to reviewer #1 suggest. |
Figure 3 , 4, 5, 6, and 17 are in different styles and colors and do not have units ( e.g. wave length) |
- These figures replaced to be 2, 3, 4, 5 and 16, however, these figures have been redrawn to be more clearer. |
For better understanding of nano tube separation, this section needs to include the analysis of the surface Zeta potential under different conditions (pH or concentrations) |
- The authors agree with this valuable suggestion. While unfortunately, the amount of available material is not sufficient to do such an analysis. Moreover, it is difficult to do this analysis because of the Covid-19 coronavirus pandemic. While this analysis will be considered in the next work. |
All the references need to be revised (e.g. 1,14, 16, 18, many more) and some need to be updated (21, 24 are too old) Reference 27 and 33 are repetitive |
- All references have been revised. - The references 21, 22, 24 were Langmuir, Freundlich, and Halsey, and these references are consider the first innovator of these models, so, it's important to mentioned. - Reference 33 was replaced to 42. |
We would like to extend our sincere thanks and appreciation to the anonymous reviewer #3. In fact, the reviewer comments, suggests, and guidance added important value to the manuscript and increased its scientific content. Therefore, the words cannot express their gratitude for their time and effort they put into evaluating this research.

Round 2
Reviewer 2 Report
The authors have improved the paper based on the comments.
Author Response
The authors would like to extend their sincere thanks and appreciation to the reviewer. In fact, their comments and guidance added a lot to the research and increased its scientific content. In the current final version, the authors considered all comments and improved the manuscript by native English Speaker
Best Regards,
Authors
Reviewer 3 Report
This study is about removing Anionic dyes with hybrid Nano tubes materials. The authors did a great work. Compared to the first draft, the manuscript structure, explanations, and references are improved. But the English writing is still needed to be improved.
Author Response

(The authors gave the same response as above.)
